# Bicrystallography and Beyond: Example of Group–Subgroup Phase Transformations

**Denis Gratias * and Marianne Quiquandon**

CNRS UMR 8247, Institut de Recherche de Chimie Paris, ENSCP, 11 rue Pierre et Marie Curie,
F-75005 Paris, France; marianne.quiquandon@chimieparistech.psl.eu
* Correspondence: denis.gratias@chimieparistech.psl.eu

**Abstract:** This paper presents the basic elementary tools for describing the global symmetry obtained by overlapping two or more crystal variants of the same structure, differently oriented and displaced one with respect to the other. It gives an explicit simple link between the concepts used in the symmetry studies on grain boundaries on one side and group–subgroup transformations on the other side. These questions are essentially of the same nature and boil down to the resolution of the same problem: identifying the permutation groups that are images of the corresponding applications. Examples are given from both domains, classical grain boundaries with coincidence lattices and group–subgroup phase transformations that illustrate the profound similarities between the two approaches.

**Keywords:** interfaces; bicrystal; *N*-crystal; group action theory; permutations homorphisms

## 1. Introduction

Understanding how geometric order can propagate in solids, for example with the generation of fascinating Moiré patterns in the superimposition of two or more identical twisted lattices, the specific relative orientations between twinned crystals or the distribution of ordered variants in the morphology of an initial single crystal after a phase transition, is a major part of describing the properties of coherent interfaces in crystals. The subject has met a particular revival since the discovery [1] of bilayers of twisted extended 2D structure like graphene and similar structures with very interesting electronic properties of the bilayer depending crucially on its symmetry. Our intention here is to give some general theoretical elements of an answer to this fascinating symmetry problem.

The notion of *bicrystal* is issued from the studies of grain boundaries in metals and minerals as first discussed by Bollman [2] after the seminal approach of Friedel on the symmetries of twinned crystals (see for instance [3,4]). First designated as dichromatic pattern (see Pond and Bollman [5] and Pond & Vlachavas [6]), the notion of bicrystal was implicitly based on the lattice symmetry mostly because examples at that time were specific studies of grain boundaries of simple metals the atomic structure of which can be described by symmorphic space groups with one unique atom per unit cell. This lack of generality has been discussed by Gratias & Portier [7] who used space groups and group action theory [8] to construct a general crystallographic framework for describing the geometry of homophase grain boundaries. In all these approaches, the basic idea was to decipher which symmetries appear in the abstract construction of the superimposition of two crystals of the same nature in different orientations for finding which elementary domain in space should be examined to discuss all the possible interfaces—grain boundaries or twins—that can actually exist between the two crystals.

The overall discussion addressed in the present paper takes its roots on the works of Guymont et al. [9] and Gratias & Portier [7] giving the basic tools for handling the symmetry properties of bicrystals using elementary group action theory [8]. Our present goal is to generalize this notion to

that of *N*-crystals, an abstract construct built of the superimposition of *N* identical crystals displaced and disoriented with respect to each other. This generalization allows the inclusion of the description of symmetry breaking in solids due to group–subgroup phase transformations that generate an assembly of crystalline domains of the low-symmetry phase out of a single crystal of the high-symmetry phase. Classical example of these phase transformation are the many chemical order–disorder transitions in metallic alloys. This leads to a complete geometric description of the interconnection between the variants generated by a group–subgroup phase transition.

The paper has three main sections:

- the first section "Standard bicrystallography" discusses the basic tools used to characterize interfaces in homogeneous crystals; a special attention is given to the building of the space group $\mathscr{W}_\alpha$ that define all possible space groups $\mathscr{P}_\tau$ of the bicrystal as function of the rigid-body translation $\tau$; another key point here is to emphasize the fact that the symmetry group of the bicrystal results from only two kind of symmetry operations these that keep both crystals simultaneously invariant and those that exchange the crystals: there is an homomorphism between the group of the bicrystal and the permutation group of two elements;
- the second section shows how to generalize the bicrystal concept to the *N*-crystal one by constructing all sets of symmetry operations, if any, that realize one of the permutations of *N* objects;
- the third and last section proposes a direct application of the notion of *N*-crystal in the case of the group–subgroup transformations.

To make the reading easy, the following notations are used all over the paper:

- point groups are designated by uppercase letters as $G$ or $H_\alpha$; point symmetry elements are designated by lowercase letters as $g$ or $\alpha$;
- space groups, including translation groups, are noted by calligraphic letters like $\mathscr{G}$ or $\mathscr{U}_\alpha$ and space symmetry elements by $\widehat{g}$ or $\widehat{\alpha}$;
- for simplicity, the term *lattice* is indiscriminately used as a set of vectors defining the lattice nodes $\lambda = na + mb + pc$, $n, m, p \in \mathbb{Z}$ or as a set of symmetry elements $(1|\lambda)$ belonging to a translation group;
- the space group of the crystal is $\mathscr{G}$ with point group (symmetry class) $G$ and lattice $\Lambda$.

## 2. Standard Bicrystallography

The crystal is characterized by its space group $\mathscr{G}$ of lattice $\Lambda$ as defined in the International Tables for Crystallography [10,11]. The space group is represented by the set of symmetry elements $\widehat{g} = (g|t)$ defined by a point symmetry operation $g$ (rotation, mirror, inversion, ...) associated with a translation $t$:

$$\widehat{g}\, r = (g|t)\, r = g\, r + t$$

The elements of $\mathscr{G}$ with the identity as point symmetry operation define the lattice $\Lambda = \{(1|\lambda) \in \mathscr{G}\}$.

The geometric operation that transforms crystal I into crystal II is an isometric transformation that takes the same form as the symmetry elements of the crystal $\widehat{\alpha} = (\alpha|\tau)$ where $\alpha$ is the point operation (rotation, mirror, inversion, ...) and $\tau$ a rigid-body translation: $\widehat{\alpha}r = (\alpha|\tau)r = \alpha r + \tau$.

Because of the intrinsic internal symmetries of the crystal, a generic point $r_1$ has infinitely many equivalent points $\mathscr{G}r_1$, called its $\mathscr{G}$-orbit, that can as well be chosen for defining $\widehat{\alpha}$; the same applies in the second crystal. Thus, the chain of equivalences:

$$r_I \xrightarrow[I]{\mathscr{G}} \mathscr{G}\, r_I \xrightarrow[I \to II]{\widehat{\alpha}} \widehat{\alpha}\mathscr{G}\, r_I \xrightarrow[II]{\widehat{\alpha}\mathscr{G}\widehat{\alpha}^{-1}} \widehat{\alpha}\mathscr{G}\widehat{\alpha}^{-1}\widehat{\alpha}\mathscr{G}\, r_I \equiv \widehat{\alpha}\mathscr{G}r_I$$

shows that the most general transformation from crystal I to II, is equivalently described by any element of the left coset $\widehat{\alpha}\mathscr{G}$ (see for instance Guymont et al. [9]):

$$r_I \xrightarrow{\widehat{\alpha}\mathcal{G}} r_{II} = \widehat{\alpha}\mathcal{G}r_I$$

The inverse transformation from II to I is characterized by the right coset $\mathcal{G}\,\widehat{\alpha}^{-1}$ as shown in Figure 1.

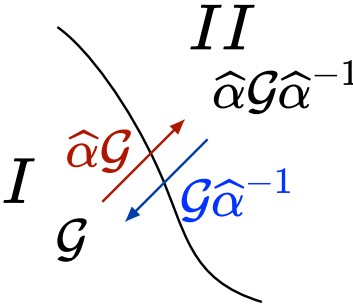

**Figure 1.** Passing from crystal I of space group $\mathcal{G}$ to crystal II of space group $\widehat{\alpha}\,\mathcal{G}\,\widehat{\alpha}^{-1}$ is achieved using any element of the left coset $\widehat{\alpha}\,\mathcal{G}$; the inverse transformation from crystal II to I is described by using any element of the right coset $\mathcal{G}\,\widehat{\alpha}^{-1}$.

### 2.1. Translation Invariants

For a generic interface, the lattices $\Lambda$ and $\Lambda'$ of respectively crystal *I* and *II* have no translation in common in the sense that no translation of crystal *II* can be expressed as a *linear combination with integer coefficients* of the basic vectors of crystal I.

There are specific cases however, like coincidence grain boundaries and twins, where the two crystals share a same translation subset of rank 3 of $\Lambda$, called the coincidence lattice (see [12]) and noted $\mathcal{T}_\alpha$, which is defined by:

$$\mathcal{T}_\alpha = \{(1|T) \in \Lambda : \exists\,(1|T') \in \Lambda \text{ such that } T' = \alpha T\} \tag{1}$$

This coincidence lattice $\mathcal{T}_\alpha$ is a subgroup of order $\Sigma$ of $\Lambda$ and $\Lambda' = \alpha\,\Lambda\,\alpha^{-1}$ and is better defined as the intersection of these two lattices:

$$\mathcal{T}_\alpha = \Lambda \,\cap\, \alpha\,\Lambda\,\alpha^{-1}$$

The second important translation group is the group $\mathcal{U}_\alpha$ generated by the union:

$$\mathcal{U}_\alpha = \Lambda \,\cup\, \alpha\,\Lambda\,\alpha^{-1}$$

This group is a supergroup of order $\Sigma$ of both $\Lambda$ and $\Lambda'$ and is also independent of the rigid-body translation. It has often been called the *Displacement Shift Complete* lattice but is now better defined as *Displacement Symmetry Conserving* lattice (see [5]) with the same acronym DSC.

The translation group $\mathcal{U}_\alpha$ is the invariance translation group of the rigid-body translation $\tau$. The rigid-body translation is indeed defined up to any translation of crystal *I* and any translation of crystal *II*: for a given point operation $\alpha$, any rigid-body translation $\tau'$ deduced from $\tau$ by a translation of $\mathcal{U}_\alpha$ leads to an equivalent bicrystal.

The translation groups $\mathcal{T}_\alpha$ and $\mathcal{U}_\alpha$ have their image in reciprocal space as demonstrated by Grimmer [13]: the intersection of the reciprocal lattices $\Lambda^*$ and $\Lambda'^* = \alpha\,\Lambda^*\,\alpha^{-1}$ is the reciprocal lattice of $\mathcal{U}_\alpha$ : $\mathcal{U}_\alpha^* = \Lambda^* \cap\, \alpha\,\Lambda^*\,\alpha^{-1}$ whereas the union group is the reciprocal lattice of $\mathcal{T}_\alpha$. These general group relations are shown in Figure 2.

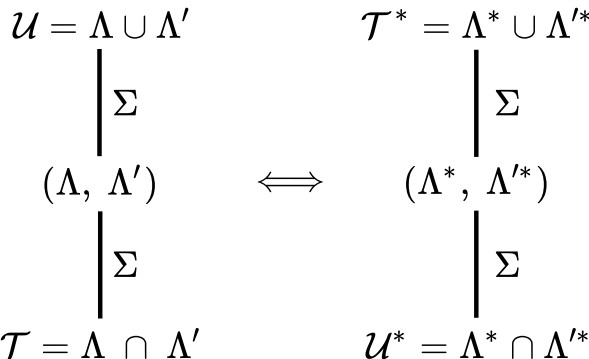

**Figure 2.** Tree of the partial order of the translation groups $\mathscr{T}_\alpha$, $\Lambda$, $\Lambda' = \alpha \Lambda \alpha^{-1}$ and $\mathscr{U}_\alpha$ in direct and reciprocal spaces (see [13]). The order sequence goes from $\mathscr{T}_\alpha$ being a subgroup of $\Lambda$ ($\Lambda'$) of order $\Sigma$ and $\Lambda$ ($\Lambda'$) being a subgroup of $\mathscr{U}_\alpha$ of same order. This corresponds in reciprocal space to exchanging the roles of $\mathscr{T}_\alpha$ and $\mathscr{U}_\alpha$ : $\mathscr{T}_\alpha^*$ is the union of the reciprocal lattices and $\mathscr{U}_\alpha^*$ is the intersection of the reciprocal lattices.

### 2.2. Symmetry and Rigid-Body Translation

Once a specific operation $\alpha$ generating a coincidence lattice is given, the question arises of which kind of space groups $\mathscr{P}_\tau$ can be found depending on the value of the associated rigid-body translation $\tau$. For that purpose, we will first determine the space group $\mathscr{W}_\alpha$ that generates the orbit of all equivalent of $\tau$ whatever its value: as will be discussed next, the little group of $\tau$ in $\mathscr{W}_\alpha$ will thus generate the space group $\mathscr{P}_\tau$.

We have seen in the preceding section that the invariant translation group of $\tau$, whatever its value, is the union group $\mathscr{U}_\alpha$ since $\tau$ is defined up to any translation of either crystals *I* or *II*. Concerning the orientational symmetries and since $\tau$ is a global translation of one of the crystals with respect to the other, only the point symmetry operations must be considered.

We use the general rule that *determining the global symmetry of the union of two equivalent objects (their normalizer) is made in searching for the symmetry elements that are common to both objects and the possible additional external elements, if any, that exchange these two objects.*

Let us designate by $G$ the point group of $\mathscr{G}$ and let $\alpha G$ be the (point) coset representing the transformation of *I* to *II*. We designate by:

$$H_\alpha = G \cap \alpha G \alpha^{-1} \text{ or } H(\alpha) = \{g \in G \text{ such that } \exists g' \in G \text{ with } \alpha g = g' \alpha\} \tag{2}$$

the point group intersection of the points groups of crystals *I* and *II*. This point group, keeping both I and II invariant, is part of the construction of $\mathscr{W}_\alpha(\alpha)$.

On the other hand, the elements common to $\alpha G$ and $G\alpha^{-1}$,

$$E_\alpha = \alpha G \cap G\alpha^{-1} \text{ or } E(\alpha) = \{\alpha g \in \alpha G \text{ such that } \exists g' \in G \text{ with } \alpha g = g' \alpha^{-1}\} \tag{3}$$

transform crystal *I* into *II* and vice-versa, generate other equivalent of $\tau$, *but in flipping it in the opposite direction $\tau \to -\tau$.* This set, if not empty, is also part of the construction of $\mathscr{W}_\alpha$.

The union:

$$W_\alpha = H_\alpha \cup E_\alpha$$

is a (point) group with $H_\alpha \lhd W_\alpha$:

i. the right and left products of an element of $H_\alpha$ with an element of $E_\alpha$ is an element of $E_\alpha$; let $h_1 \in H_\alpha$, $\alpha h_2 \in E_\alpha$, then:

$$h_1(\alpha h_2) = h_1 h_2' \alpha^{-1} \in G\alpha^{-1} \text{ and } h_1(\alpha h_2) = \alpha h_1' h_2 \in \alpha G$$
$$(\alpha h_2)h_1 = h_2' \alpha^{-1} \alpha h_1' \widehat{\alpha}^{-1} \in G\alpha^{-1} \text{ and } (\alpha h_2)h_1 = \alpha h_2 h_1 \in \alpha G$$

ii.　the product of two elements of $E_\alpha$ is an element of $H_\alpha$; let $\alpha h_1, \alpha h_2 \in E_\alpha$, then:

$$(\alpha h_1)(\alpha h_2) = \alpha h_1 h_2' \alpha^{-1} \in \alpha G \alpha^{-1} \text{ and } (\alpha h_1)(\alpha h_2) = h_1' \alpha^{-1} \alpha h_2 = h_1' h_2 \in G$$

Therefore, the group $W_\alpha$ is either $H_\alpha$ itself if $E_\alpha$ is empty, or a supergroup of order 2 of $H_\alpha$ if $E_\alpha$ is not empty; in that later case, it can be written as:

$$W_\alpha = H_\alpha \cup \epsilon H_\alpha$$

with $\epsilon \in E_\alpha$.

It is now trivial to find the group $\mathscr{W}_\alpha$ describing all the possible symmetries of the bicrystals generated by a given point operation $\alpha$ according to the rigid-body translation $\tau$; it is given by the direct product of $\mathscr{W}_\alpha$ with $\mathscr{U}_\alpha$:

$$\mathscr{W}_\alpha = (H_\alpha \cup \epsilon H_\alpha) \times \mathscr{U}_\alpha.$$

This space group $\mathscr{W}_\alpha$ contains at once all the symmetry information for any value of $\tau$. It is enough to restrict $\tau$ inside the Dirichlet domain of $\mathscr{W}_\alpha$ to find all the possible space groups $\mathscr{P}_\tau$ depending on $\tau$ using the following properties:

- the number of different possible types of space groups $\mathscr{P}_\tau$ of the bicrystal created by a point operator $\alpha$ and a rigid-body translation $\tau$ is the number of different position strata (see [14,15]) of the space group $\mathscr{W}_\alpha$ that $\tau$ can take;
- the space group $\mathscr{P}_\tau$ of the bicrystal is determined by the point group of the symmetry stratum of $\tau$ in $\mathscr{W}_\alpha$; in particular, high-symmetry groups appear when $\tau$ points on special positions (strata of dimension 0) of $\mathscr{W}_\alpha$; they correspond to the so-called symmetry dictated extrema as discussed by Cahn & Kalonji [16];
- in the construction of $\mathscr{P}_\tau$, the generating elements belonging to the intersection group $H_\alpha$ are to be taken as they are, whereas those corresponding to the coset $\epsilon H_\alpha$ *must be multiplied by the inversion* before being injected in $\mathscr{P}_\tau$ (for example, a mirror in the coset $\epsilon H_\alpha$ generates a two-fold axis perpendicular to the mirror as generator of $\mathscr{P}_\tau$).

*2.3. The Space Group of the Bicrystal*

Calculating $\mathscr{P}_\tau$ is similar to the previous derivation in using space groups instead of point groups. Here again, the space group $\mathscr{P}_\tau$ is constituted of two kinds of space symmetry elements:

- those that leave invariant simultaneously $I$ and $II$: they form a subgroup of $\mathscr{G}$ noted $\mathscr{I}_\alpha$ and defined by $\mathscr{I}_\alpha = \mathscr{G} \cap \widehat{\alpha} \, \mathscr{G} \, \widehat{\alpha}^{-1}$;
- those that exchange $I$ and $II$: $\mathscr{E}_\alpha = \widehat{\alpha} \, \mathscr{G} \cap \mathscr{G} \, \widehat{\alpha}^{-1}$ that are outside $\mathscr{G}$ and can reduce to the empty set.

The elements (For example, the subgroup $\mathscr{G}_\alpha$ of $\mathscr{G}$ defined by the elements of $\mathscr{G}$ that commute with $\widehat{\alpha}$, $\mathscr{G}_\alpha = \{\widehat{g} \in \mathscr{G} \text{ such that } \widehat{g}\,\widehat{\alpha} = \widehat{\alpha}\,\widehat{g}\}$ is a subgroup of $\mathscr{I}_\alpha$. This is the case when $\widehat{\alpha}$ is a pure rotation sharing the same axis as a pure rotation $\widehat{g}$ of $\mathscr{G}$, then $\widehat{g}$ is in $\mathscr{I}_\alpha$) of $\mathscr{I}_\alpha$ are defined by:

$$\mathscr{I}_\alpha = \{\widehat{g} \in \mathscr{G} : \exists \, \widehat{g}' \in \mathscr{G} \text{ such that } \widehat{g}' = \widehat{\alpha}\widehat{g}\widehat{\alpha}^{-1} \text{ or } \widehat{\alpha}\widehat{g} = \widehat{g}'\widehat{\alpha}\}$$

The elements (For example, the binary elements of $\widehat{\alpha} \, \mathscr{G}$ modulo $\mathscr{G}$ are elements of $\mathscr{E}_\alpha$: let $(\widehat{\alpha} \, \widehat{g}_1)\,(\widehat{\alpha} \, \widehat{g}_1) = \widehat{g}_2$, then:

$$\widehat{\alpha} \, \widehat{g}_1 = (\widehat{\alpha} \, \widehat{g}_1)(\widehat{\alpha} \, \widehat{g}_1) \, \widehat{g}_1^{-1}\widehat{\alpha}^{-1} = \widehat{g}_2 \, \widehat{g}_1^{-1}\widehat{\alpha}^{-1} \in \mathscr{G} \, \widehat{\alpha}^{-1}.$$

As an example, the product $\hat{\alpha}\,\hat{m}$ of a rotation of angle $\alpha$ around an axis contained in the plane of a mirror $\hat{m}$ element of $\mathscr{G}$ is an element of $\mathscr{E}_\alpha$ : it generates a mirror of $\mathscr{P}_\tau$ the plane of which is the mirror plane of $\hat{m}$ rotated by $\alpha/2$) of $\mathscr{E}_\alpha$ — if not empty — are defined by:

$$\mathscr{E}_\alpha = \{\widehat{\alpha\hat{g}},\ \hat{g} \in \mathscr{G} : \exists\ \hat{g}' \in \mathscr{G} \text{ such that } \widehat{\alpha\hat{g}} = \hat{g}'\hat{\alpha}^{-1}\}$$

As in the preceding section, it is easily demonstrated that the union of the two sets $\mathscr{I}_\alpha$ and $\mathscr{E}_\alpha$ (if not empty) forms a group $\mathscr{P}_\tau$:

$$\mathscr{P}_\tau = \mathscr{I}_\alpha \cup \mathscr{E}_\alpha = \mathscr{I}_\alpha \cup \hat{\epsilon}\ \mathscr{I}_\alpha, \quad \hat{\epsilon} \in \mathscr{E}_\alpha,\ \mathscr{I}_\alpha \lhd \mathscr{P}_\tau \tag{4}$$

This space group of the bicrystal with translation group $\mathscr{T}_\alpha$ is also designated as the symmetry group of the *dichromatic pattern* (see [17]).

The explicit computation of $\mathscr{P}_\tau$ is made particularly simple in considering the little group of $\tau$ in $\mathscr{W}_\alpha$:

- the elements of $\mathscr{I}_\alpha$ are such that:

$$(g|t)(\alpha|\tau) = (\alpha|\tau)(g'|t') \text{ or } (g\alpha|g\tau + t) = (\alpha g'|\alpha t' + \tau)$$

  leading to $g\alpha = \alpha g'$ and $g\tau - \tau = \alpha t' - t$; this is achieved for $g$ belonging to the little group of $\tau$ ($g\tau = \tau$) where $t$ and $t'$ are translations of $\mathscr{T}_\alpha$ ($\alpha t' = t$);
- the elements of $\mathscr{E}_\alpha$ are such that:

$$(g|t)(\alpha|\tau)^{-1} = (\alpha|\tau)(g'|t') \text{ or } (g\alpha^{-1}|t - g\alpha^{-1}\tau) = (\alpha g'|\alpha t' + \tau)$$

  leading to $g\alpha^{-1} = \alpha g'$ and $g\alpha^{-1}\tau + \tau = -\alpha t' + t$; this is achieved for $g\alpha^{-1}$ being element of the little group of $\tau$ *multiplied by the inversion* ($g\alpha^{-1}\tau = -\tau$) where $t$ and $t'$ are translations of $\mathscr{T}_\alpha$ ($\alpha t' = t$).

This shows that the little (point) group of $\tau$ in $\mathscr{W}_\alpha$, say $W_\tau = H_\tau \cup \epsilon_\tau H_\tau$ is directly connected to the point group $P_\tau$ of the space group $\mathscr{P}_\tau$ that is built with $H_\tau$ and $\epsilon_\tau H_\tau$ multiplied by the inversion:

$$P_\tau = H_\tau \cup \bar{1}\epsilon_\tau H_\tau \tag{5}$$

The number of different strata of $\mathscr{W}_\alpha$ gives the number of various kind of space groups $\mathscr{P}_\tau$ depending on the values of $\tau$. In particular, the symmetry dictated values of $\tau$ are those corresponding to strata of dimension 0 of $\mathscr{W}_\alpha$.

### 2.4. A Simple 2D Example

To exemplify these rules, we consider the classical case of the bicrystal generated by the rotation of a 2D square structure (see for instance [18–21] ) onto itself as shown in Figure 3. The actual structure belongs to either symmetry class $G = 4$ (space group $\mathscr{G} = p4$) or $G = 4mm$ that is the holohedry of the square system (space groups $\mathscr{G} = p4mm$ and $p4gm$). Coincidence lattices appear for rotations $\alpha = 2\arctan(m/n)$ around the origin that superimpose the node $(n, -m)$ on top of the node $(n, m)$, with $n$, $m$ coprime and $0 < m < n \in \mathbb{Z}$. This leads to the coincidence lattice defined by the vectors $T_1 = (n, m)$, $T_2 = (-m, n)$; the index is $\Sigma = n^2 + m^2$ and the union lattice $\mathscr{U}_\alpha$ is defined by $U_1 = T_1/\Sigma$, $U_2 = T_2/\Sigma$.

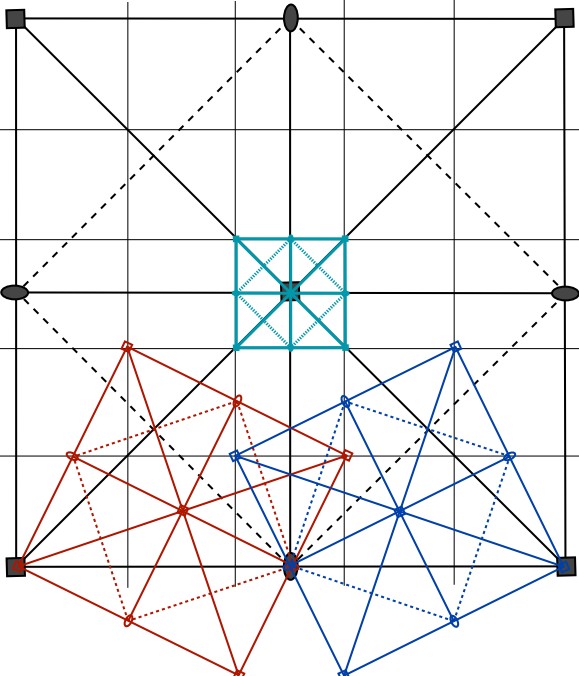

**Figure 3.** Example of two square crystals of space groups $p4mm(\Lambda)$ rotated by $\alpha = 53.1301°$, drawn in red and blue, leading to a coincidence lattice $T_1 = (2,1)$, $T_2 = (-1,2)$, $\Sigma = 5$, in black with same square symmetry $p4mm(T_1, T_2)$. The space group $\mathscr{W}_\alpha = p4mm(U_1, U_2)$ is shown in turquoise with $U_1 = T_1/5$ and $U_2 = T_2/5$. The number of various possible space groups of the (red-blue) bicrystals according to the value of the rigid-body translation are the number of position strata of $p4mm(U_1, U_2)$ as given in Table 1.

The computation of $\mathscr{W}_\alpha(\alpha)$ is quite simple. The intersection $H_\alpha$ is the set

$$H(\alpha) = \{g \in G \text{ such that } g\alpha = \alpha g\} = 4$$

and the exchange set is the set of the mirrors in $\alpha G \cap G\alpha^{-1}$ corresponding to the initial mirrors, $m_x(m_y)$, $m_{xy}(m_{x\bar{y}})$ rotated by $\alpha/2$ for the class $4m$. This set is empty for the case of the class 4. Therefore:

$$\text{Class } 4mm : \mathscr{W}_\alpha(\alpha) = p4mm(U_1, U_2), \quad \text{Class } 4 : \mathscr{W}_\alpha(\alpha) = p4(U_1, U_2).$$

This demonstrated that *whatever the value* of the rotation $\alpha$ generating a coincidence lattice, the bicrystal of a $p4$ structure can have only four space groups: $p4(\times 2)$, $p2$ and $p1$ corresponding to the four strata of the group $p4$, whereas structures of space groups $p4mm$ and $p4gm$ have seven possible space groups from $p4mm$ to $p1$ corresponding to the seven strata of the group $p4mm$ as shown on Table 1 and exemplified in Figure 4 for the values of $\tau$ corresponding to the special points.

Here, in the holohedral examples, all 4-fold and 2-fold rotations belong to $H_\alpha$ whereas the exchange elements are the original mirrors rotated by $\alpha/2$. It is interesting to observe in the case of the strata of dimension 1 in Table 1, that, as expected, the actual mirrors of $\mathscr{P}_\tau$ are perpendicular to those of the little group of $\tau$ as expected in relation (5); for instance a little group $m_x$ generates a space group $pm_y$ or $pg_y$ and a little group $m_{xy}$ generates a mirror $m_{x\bar{y}}$.

**Table 1.** The complete set of symmetry groups $\mathscr{P}_\tau$ for general square structures of space groups $p4mm$, $p4gm$ and $p4$ as function of the rigid-body translation $\tau$. Structures of groups $p4mm$ and $p4gm$ share the same $\mathscr{W}_\alpha = p4mm$ and structures of group $p4$, generating no exchange set, have $\mathscr{W}_\alpha = p4$. The coordinates of the translation $\tau$ are given with respect to the unit cell of the union group $\mathscr{U}_\alpha$ defined by $U_1 = (n, m)/\Sigma$ and $U_2 = (-m, n)/\Sigma$ with $\Sigma = n^2 + m^2$. The (primitive) unit cell of the groups $\mathscr{P}_\tau$ are defined by $T_1 = (n, m)$ and $T_2 = (-m, n)$.

| $\tau$ | Stratum Dim | Little Group in $\mathscr{W}_\alpha$ | | $\mathscr{P}_\tau$ for $p4mm$ | $\mathscr{P}_\tau$ for $p4gm$ | $\mathscr{P}_\tau$ for $p4$ |
|---|---|---|---|---|---|---|
| $(0, 0)$ | 0 | $4mm$ | 4 | $p4mm$ | $p4gm$ | $p4$ |
| $(1/2, 1/2)$ | 0 | $4mm$ | 4 | $p4gm$ | $p4mm$ | $p4$ |
| $(1/2, 0)$ | 0 | $2mm$ | 2 | $p2mg$ | $p2gm$ | $p2$ |
| $(x, y)$ | 2 | $1$ | | $p1$ | $p1$ | $p1$ |
| $(x, 0)$ | 1 | $.m_x.$ | | $pm_y$ | $pg_y$ | — |
| $(x, 1/2)$ | 1 | $.m_x.$ | | $pg_y$ | $pm_y$ | — |
| $(x, x)$ | 1 | $..m_{x\bar{y}}$ | | $cm_{xy}$ | $cm_{xy}$ | — |

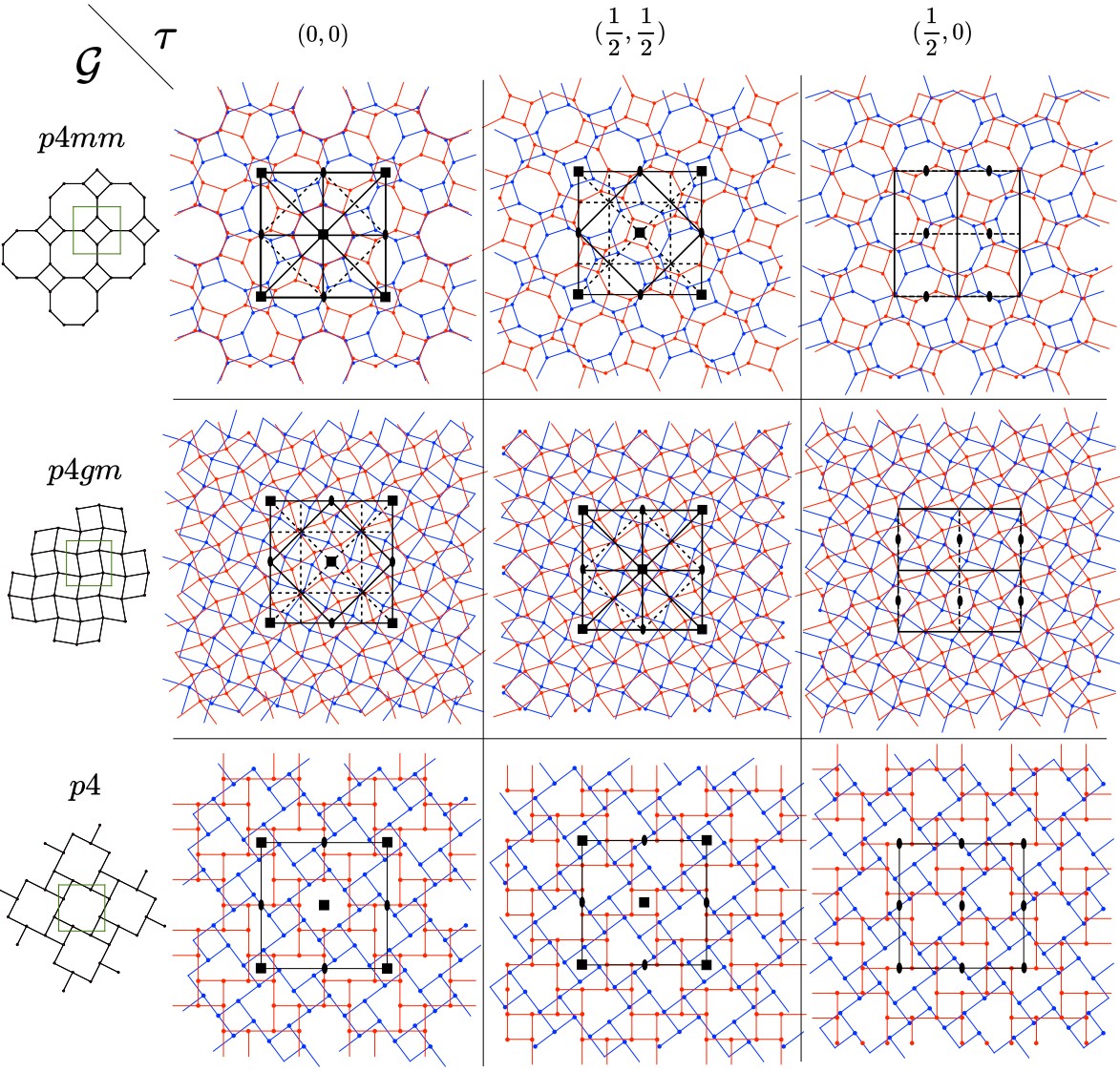

**Figure 4.** The possible space groups $\mathscr{P}_\tau$ given in Table 1 for the bicrystal generated by the rotation $\alpha = 2 \arctan(m/n)$ (here for $n = 2$, $m = 1$, $\Sigma = 5$, $\alpha = 53.1301°$) as a function of the rigid-body translation $\tau$ at the special points $(0, 0)$, $(1/2, 1/2)$ and $(1/2, 0)$ for the three square symmetry space groups $p4mm$, $p4gm$ and $p4$. This scheme holds for any integer values of $n$ and $m$.

### 2.5. The Special Case of General Quasi-Bicrystals

The physical meaning of the rigid-body translation $\tau$ deserves special attention for the case of general crystal superimposition with no coincidence lattice. This case corresponds to the limit $\Sigma \to \infty$ where the bicrystal becomes a quasiperiodic structure (see for instance [22,23]). The translation group $\mathscr{U}_\alpha$ becomes thus a $\mathbb{Z}-$module and the size of the unit cell of $\mathscr{U}_\alpha$ reduces to zero. We call it a *quasi-bicrystal*. This means that the general quasi-bicrystal is then independent of $\tau$: any two quasi-bicrystals built with the same *I* and *II* crystals that differ only by a change $\delta\tau$ in the rigid-body translation are locally isomorphic, i.e., any local configuration of finite size in one quasi-bicrystal is present in the other with the same frequency and vice-versa. The two quasi-bicrystals are thus *undistinguishable* in the sense that modifying the rigid-body translation between the two crystals has the only effect of redistributing in space the atomic configurations of finite size with the same frequencies without adding any new configurations nor removing any existing ones.

## 3. From Bicrystallography to $N$-Crystallography

We can extend the *bicrystal* concept to that of $N-crystal$ where $N > 2$ crystals are superimposed in observing that, for $\mathscr{E}$ non-empty, the quotient group $\mathscr{P} \,/\, \mathscr{I}$ is isomorphic to the permutation group of two objects $\mathscr{S}_2$:

$$\mathscr{P} \,/\, \mathscr{I} = \mathscr{S}_2. \tag{6}$$

This basic isomorphism is the starting point for making an easy generalization to $N$-crystal.

### 3.1. The N-Crystal Generalization

The overall symmetry group of $N$ equivalent objects for finite $N$, is isomorphic to a subgroup of the permutation group $\mathscr{S}_N$ of $N$ objects. Thus, constructing the symmetry of the abstract $N-crystal$, consists of identifying which actual symmetry elements are representative of those of $\mathscr{S}_N$.

For notation coherency, we note the identity as $\widehat{\alpha}_1 = Id$. The elements of $\mathscr{S}_N$, say $\pi_k$, transform the $N$-uplet of integers $(1, 2, \ldots, N)$ into $(\pi_k(1), \pi_k(2), \ldots, \pi_k(N))$. On the other hand, transforming crystal $i$ into $j$ is achieved by the elements of the coset (if not empty) $\widehat{\alpha}_j \, \mathscr{G} \, \widehat{\alpha}_i^{-1}$ as illustrated in Figure 5.

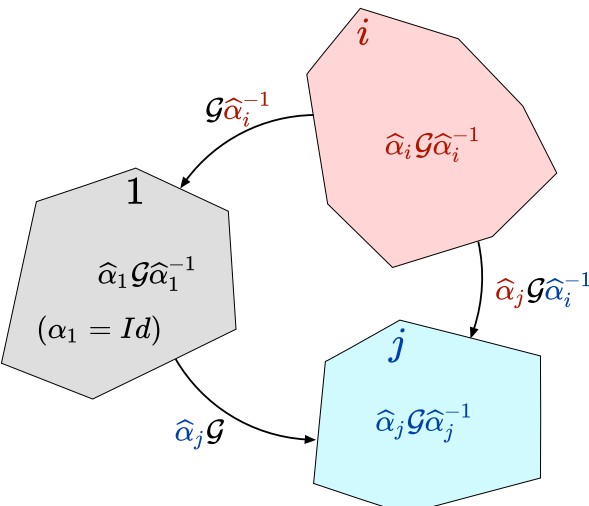

**Figure 5.** Transforming variant $i$ into $j$ in a $N$-crystal is achieved by transforming $i$ back into the reference variant noted 1 using $\mathscr{G}\widehat{\alpha}_i^{-1}$ and then into variant $j$ by applying $\widehat{\alpha}_j\mathscr{G}$. This leads to the global operation $\widehat{\alpha}_j \, \mathscr{G} \, \widehat{\alpha}_i^{-1}$. Then, the transposition $(i, j) \to (j, i)$ has the image $(\widehat{\alpha}_j \, \mathscr{G} \, \widehat{\alpha}_i^{-1}) \cap (\widehat{\alpha}_i \, \mathscr{G} \, \widehat{\alpha}_j^{-1})$ (possibly empty) in the standard crystallographic representation.

At the substitution $j \to p_k(j)$, corresponds the coset $\widehat{\boldsymbol{\alpha}}_{p_k(j)} \mathscr{G} \widehat{\boldsymbol{\alpha}}_j^{-1}$. Thus, at the permutation $p_k$ of $N$ objects:

$$(1, 2, \ldots, j, \ldots, N) \to (p_k(1), p_k(2), \ldots, p_k(j), \ldots, p_k(N))$$

corresponds the intersection (if not empty) of the cosets $\widehat{\boldsymbol{\alpha}}_{p_k(j)} \mathscr{G} \widehat{\boldsymbol{\alpha}}_j^{-1}$ for all $j$ varying from 1 to $N$:

$$(p_k(1), p_k(2), \ldots, p_k(j), \ldots, p_k(N)) \implies \cap_{j=1}^N \widehat{\boldsymbol{\alpha}}_{p_k(j)} \mathscr{G} \widehat{\boldsymbol{\alpha}}_j^{-1}$$

The symmetry group $\mathscr{P}$ of the $N$-crystal is the union of the $N!$ such intersections (many of them can be empty) associated with the $N!$ permutations of the group $\mathscr{S}_n$:

$$\mathscr{P} = \cup_{k=1}^{N!} \left( \cap_{j=1}^N \widehat{\boldsymbol{\alpha}}_{p_k(j)} \mathscr{G} \widehat{\boldsymbol{\alpha}}_j^{-1} \right) \tag{7}$$

The mapping between each element of the permutation group and its corresponding coset intersections (possibly empty) insures $\mathscr{P}$ to be a group.

Introducing the invariant subgroup $\mathscr{I} \lhd \mathscr{P}$ generated by the identity permutation $p_1(j) = j$:

$$\mathscr{I} = \cap_{j=1}^N \widehat{\boldsymbol{\alpha}}_j \mathscr{G} \widehat{\boldsymbol{\alpha}}_j^{-1},$$

we find the basic decomposition for the $N$-crystal symmetry group:

$$\mathscr{P} = \cup_{k=1}^{N!} \varepsilon_k \mathscr{I}, \text{ with } \varepsilon_1 = Id \tag{8}$$

and

$$\mathscr{P} / \mathscr{I} \simeq \sigma_N \leq \mathscr{S}_{\mathcal{N}}$$

where $\sigma_N$ is a subgroup of $\mathscr{S}_{\mathcal{N}}$.

Expressions (7) and (8) are the explicit formula that define the symmetry group of the $N$-crystal as natural extensions of formula (4) of the bicrystal.

### 3.2. Application to Group–Subgroup Phase Transformations

The simplest and most obvious application of using relation (8) is the classical group–subgroup phase transformations as exemplified by the numerous cases encountered in solid state physics of alloys with the numerous examples of order–disorder transformations. These are characterized by a symmetry breaking at some critical temperature—or on reaching a two-phased region in the phase diagram—from a high-symmetry group $\mathscr{G}$ at high temperature to a low-symmetry subgroup $\mathscr{H}$ of order $N$ in $\mathscr{G}$ at low temperature:

$$\mathscr{G} \to [\mathscr{G} : \mathscr{H}], \quad \mathscr{H} < \mathscr{G}, \quad \mathscr{G} = \cup_{i=1}^N \widehat{\boldsymbol{g}}_i \mathscr{H}, \widehat{\boldsymbol{g}}_1 = Id$$

The decomposition of $\mathscr{G}$ in cosets of $\mathscr{H}$ corresponds to a single crystal of space group $\mathscr{G}$ at high temperature transforming, at low temperature, into numerous crystallites of phase $\mathscr{H}$ that distribute into $N$ kind of differently oriented/translated crystals $\mathscr{H}$. Each family, called a variant, gather all crystallites described by the same space group $\mathscr{H}$. The coset $\widehat{\boldsymbol{g}}_i \mathscr{H}$ is the set of all equivalent operations that transform a given chosen variant of space group $\mathscr{H}$ into another of space group $\widehat{\boldsymbol{g}}_i \mathscr{H} \widehat{\boldsymbol{g}}_i^{-1}$. Passing from the variant $i$ to the variant $j$ is achieved by the set of operations $\widehat{\boldsymbol{g}}_j \mathscr{H} \widehat{\boldsymbol{g}}_i^{-1}$. We recognize here the expressions discussed in the previous section where the operations $\widehat{\boldsymbol{\alpha}}_i$ are identified to $\widehat{\boldsymbol{g}}_i$. Therefore, the symmetry group $\mathscr{P}_\tau$ is the group $\mathscr{G}$ itself since the $N$-crystal is the high temperature phase:

$$\mathscr{G} = \mathscr{P} = \cup_{k=1}^{N!} \left( \cap_{j=1}^N \widehat{\boldsymbol{g}}_{p_k(j)} \mathscr{H} \widehat{\boldsymbol{g}}_j^{-1} \right), \widehat{\boldsymbol{g}}_1 = Id \tag{9}$$

The group $\mathscr{G}$ of the high temperature phase describes the way the $N$ variants of the low temperature phase transform into each other. We designate by $\mathscr{I}$ the invariant subgroup of order $K$ in $\mathscr{H}$ that is the intersection of the groups of the $N$ variants:

$$\mathscr{I} = \cap_{i=1}^{N} \widehat{g}_i \, \mathscr{H} \, \widehat{g}_i^{-1}$$

This intersection group is the kernel of the application $\mathscr{G} \rightarrow [\mathscr{G} : \mathscr{H}]$ of $\mathscr{G}$ on its coset decomposition on $\mathscr{H}$. The pertinent group to be considered for discussing the symmetry breaking induced by the transformation is the image of the application. It is associated with the symmetry group $\sigma_N$ of order $N \times K$, isomorphic to a subgroup of $\mathscr{S}_N$ obtained by computing the coset decomposition $\mathscr{G} = \mathscr{P}$ of $\mathscr{I}_\alpha$ and extracting its quotient by $\mathscr{I}_\alpha$:

$$\mathscr{G}(=\mathscr{P}) = \cup_{j=1}^{N \times K} \varepsilon_j \, \mathscr{I}_\alpha, \quad \sigma_n \simeq \mathscr{G} \, / \, \mathscr{I}$$

This image group $\sigma_N \simeq \mathscr{G}/\mathscr{I}$ gives the complete and faithful description of how the parent group $\mathscr{G}$ operates on the variants $\mathscr{H}$.

### 3.3. Simple Examples

#### 3.3.1. A 2D Toy-Model

We consider the hypothetical phase transition of blue atoms decomposing by chemical ordering into red, green and brown atoms as shown in Figure 6. This order–disorder transformation is characterized by the group–subgroup relation:

$$\mathscr{G} = p3m1(a, b) \rightarrow \mathscr{H} = cm1(a, c = a + 2b)$$

with standard coset decomposition $[\mathscr{G} : \mathscr{H}]$:

$$p3m1(a, b) = \{\widehat{\mathbf{1}} + \widehat{\mathbf{3}}^1 + \widehat{\mathbf{3}}^2\}cm1(a, c)$$

corresponding to generate 3 variants deduced from each other by the 3-fold rotation lost during the transition.

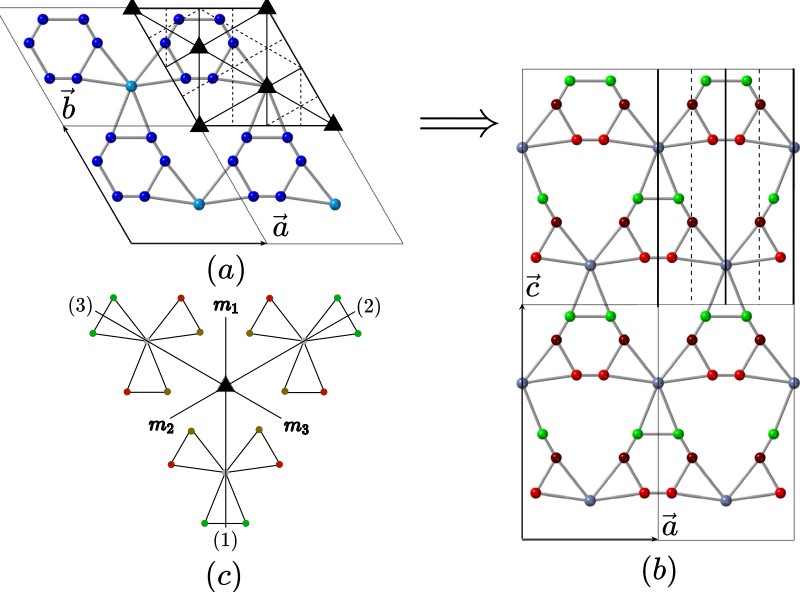

**Figure 6.** Group–subgroup phase transition from (**a**) $p3m1(a, b)$ to (**b**) $cm1(a, c = a + 2b)$ generating three variants (**c**) deduced from each other by the permutation group $\mathscr{S}_3$ isomorphic to $3m$.

We note $\widehat{g}_1 = \widehat{\mathbf{1}}$, $\widehat{g}_2 = \widehat{\mathbf{3}}^1 = (3^1|0,0)$ and $\widehat{g}_3 = \widehat{g}_2^{-1} = \widehat{\mathbf{3}}^2 = (3^2|0,0)$. The permutation group (equivalent here to a three-color symmetry group) is obtained by computing the quotient group of $\mathscr{G}$ on the kernel $\mathscr{I}$:

$$\mathscr{I} = \mathscr{H} \cap \widehat{g}_2 \mathscr{H} \, \widehat{g}_2^{-1} \cap \widehat{g}_3 \mathscr{H} \, \widehat{g}_3^{-1} = p1(a,b)$$

There are thus 6 cosets in the decomposition of $\mathscr{G}$ on $\mathscr{I}$ defined by the following variant permutations and corresponding cosets intersections:

$$\mathscr{I}_\alpha: \quad \widehat{\mathbf{1}} \quad \rightarrow (1,2,3) \longleftrightarrow \widehat{g}_1 \, \mathscr{G} \, \widehat{g}_1^{-1} \cap \widehat{g}_2 \, \mathscr{G} \, \widehat{g}_2^{-1} \cap \widehat{g}_3 \, \mathscr{G} \, \widehat{g}_3^{-1}$$

$$\mathscr{E}_\alpha: \begin{cases} \widehat{\mathbf{3}}^1 & \rightarrow (2,3,1) \longleftrightarrow \widehat{g}_2 \, \mathscr{G} \, \widehat{g}_1^{-1} \cap \widehat{g}_3 \, \mathscr{G} \, \widehat{g}_2^{-1} \cap \widehat{g}_1 \, \mathscr{G} \, \widehat{g}_3^{-1} \\ \widehat{\mathbf{3}}^2 & \rightarrow (3,1,2) \longleftrightarrow \widehat{g}_3 \, \mathscr{G} \, \widehat{g}_1^{-1} \cap \widehat{g}_2 \, \mathscr{G} \, \widehat{g}_3^{-1} \cap \widehat{g}_1 \, \mathscr{G} \, \widehat{g}_2^{-1} \\ \widehat{m}_1 & \rightarrow (1,3,2) \longleftrightarrow \widehat{g}_1 \, \mathscr{G} \, \widehat{g}_1^{-1} \cap \widehat{g}_3 \, \mathscr{G} \, \widehat{g}_2^{-1} \cap \widehat{g}_2 \, \mathscr{G} \, \widehat{g}_3^{-1} \\ \widehat{m}_2 & \rightarrow (3,2,1) \longleftrightarrow \widehat{g}_3 \, \mathscr{G} \, \widehat{g}_1^{-1} \cap \widehat{g}_2 \, \mathscr{G} \, \widehat{g}_2^{-1} \cap \widehat{g}_1 \, \mathscr{G} \, \widehat{g}_3^{-1} \\ \widehat{m}_3 & \rightarrow (2,1,3) \longleftrightarrow \widehat{g}_2 \, \mathscr{G} \, \widehat{g}_1^{-1} \cap \widehat{g}_1 \, \mathscr{G} \, \widehat{g}_2^{-1} \cap \widehat{g}_3 \, \mathscr{G} \, \widehat{g}_3^{-1} \end{cases}$$

and the explicit *3-crystal* symmetry group can be written as:

$$\mathscr{G} = \mathscr{P} = \{\widehat{\mathbf{1}} + \widehat{\mathbf{3}}^1 + \widehat{\mathbf{3}}^2 + \widehat{m}_1 + \widehat{m}_2 + \widehat{m}_3\} \, p1(a,b)$$

the image of which is the group of permutation of three objects of order 6:

$$\sigma_3 = \{(1,2,3),(2,3,1),(3,1,2),(1,3,2),(3,2,1),(2,1,3)\}$$

isomorphic to the point group $3m$. This is shown in Figure 6c where the drawing of the three-fold axis and the three mirrors $m_1$, $m_2$ and $m_3$ explicitly demonstrates how the three variants connect to each other according to $3m$.

A few other classical examples in the family of chemical ordering of metallic alloys are the following.

### 3.3.2. CuZn $B_2$ :

Here we have $\mathscr{G} = Im3m(a)$, $\mathscr{H} = Pm3m(a)$, $N = 2$, $\mathscr{I} = \mathscr{H}$, $\sigma_2 \simeq 2 = \mathscr{S}_2$

$$Im3m(a) = \{(1|000) + (1|1/2,1/2,1/2)\} Pm3m(a)$$

Here, we are in the simplest case of bicrystal induced by transformation where the translation $1/2(1,1,1)$ is the generator of the exchange set (color translation). The image group is $\mathscr{S}_2$.

### 3.3.3. Cu$_3$Au L1$_2$ :

$\mathscr{G} = Fm3m(a)$, $\mathscr{H} = Pm3m(a)$, $N = 4$, $\mathscr{I} = Pmmm(a)$, $\sigma_4 \simeq m3 = \mathscr{S}_4$

$$Fm3m(a) = \{(1|000) + (1|1/2,1/2,0) + (1|0,1/2,1/2) + (1|1/2,0,1/2)\}$$
$$\times \{(1|000) + (C_3|0,0,0) + (C_3^2|0,0,0)\} \times \{(1|000) + (m_{x,y}|000)\} Pmmm(a)$$

Here the image group is isomorphic to $m3$ (regular tetrahedron). We can represent the four variants as the vertices of a regular tetrahedron. For example applying the translation $1/2(1,1,0)$ to the four variants corresponds to the permutation $(1,2,3,4) \rightarrow (2,1,4,3)$.

## 4. Conclusions

We have shown here that the symmetry group of the superimposition of two crystals related by the boundary operation $\widehat{\alpha} = (\alpha|\tau)$ is the combination of two kind of operations: those that

leave simultaneous invariant both crystals ($\mathscr{I}_\alpha = \mathscr{G} \cap \widehat{\boldsymbol{\alpha}} \mathscr{G} \widehat{\boldsymbol{\alpha}}^{-1}$) and those that exchange them ($\varepsilon \mathscr{I}_\alpha = \widehat{\boldsymbol{\alpha}} \mathscr{G} \cap \mathscr{G} \widehat{\boldsymbol{\alpha}}^{-1}$):

$$\mathscr{P} = \mathscr{I}_\alpha \, \cup \, \varepsilon \mathscr{I}_\alpha$$

and, when the exchange set $\varepsilon \mathscr{I}_\alpha$ is non-empty, the quotient group $\mathscr{P} / \mathscr{I}_\alpha$ is isomorphic to $\mathscr{S}_2$, the permutation group of two objects.

The space group $\mathscr{P}$ depends on both the point operation $\alpha$ and the translation part $\tau$ of $\widehat{\boldsymbol{\alpha}}$. The number of different possible space groups associated with a given point operation $\alpha$ according to the $\tau$ value is equal to the number of different strata of the group:

$$\mathscr{W}_\alpha = [(G \cup \alpha G \alpha^{-1}) \cap (\alpha G \cup G \alpha^{-1})] \times \mathscr{U}_\alpha$$

where $G$ is the point group of the structure and $\mathscr{U}_\alpha$ the union group of the lattices of crystals I and II.

The symmetry group of the superimposition of a *finite number* (if $N \to \infty$, as for cubic variants generated by iterative mirror twinning on the $[1, 1, 1]$ plane, the present description fails in which case the analysis developed by Cayron (see for instance [24]) using groupoids that is to be considered), say $N$, of variants is the combination of all possible symmetry operation that permute the $N$ variants and can be written:

$$\mathscr{P} = \cup_{k=1}^{N!} \left( \cap_{j=1}^{N} \widehat{\boldsymbol{\alpha}}_{p_k(j)} \, \mathscr{G} \, \widehat{\boldsymbol{\alpha}}_j^{-1} \right)$$

where the $N$ structurally identical variants are related by $\widehat{\boldsymbol{\alpha}}_j \mathscr{G}$ cosets with $j = 1, N$. Here again, the quotient group $\sigma_N \simeq \mathscr{P} / \mathscr{I}$, where $\mathscr{I} = \cap_{j=1}^{N} \widehat{\boldsymbol{g}}_j \, \mathscr{H} \, \widehat{\boldsymbol{g}}_j^{-1}$ is a subgroup of the permutation group $\mathscr{S}_N$ of $N$ objects.

This generalization allows the gathering in a unique scheme of the symmetry studies of $N$-crystallography on special grain boundaries of $N$ (finite) variants and the coset decomposition in $N$ variants due to a group–subgroup solid state transformation.

The only problem here was to determine the mapping between the actual symmetry crystallographic elements on one side and the abstract permutation operations on the other side. As always when using group theory, the pertinent mapping is obtained by extracting the image group of the application, say $\varphi$, as the quotient of the overall symmetry group ($\mathscr{P}$ or $\mathscr{G}$) by the kernel of the application given by the intersection group $\mathscr{I}$ that is invariant in the parent group:

$$\mathscr{G} \xrightarrow{\varphi} [\mathscr{G} : \mathscr{H}], \; \mathscr{I} = \cap_{j=1}^{N} \widehat{\boldsymbol{g}}_j \, \mathscr{H} \, \widehat{\boldsymbol{g}}_j^{-1}$$
$$\mathrm{Im} \, \varphi = \mathscr{G} / \mathscr{I} \simeq \sigma_N \leq \mathscr{S}_N.$$

or equivalently:

$$\mathscr{G} \xrightarrow{\varphi} \mathscr{P} = \cup_{k=1}^{N!} \left( \cap_{j=1}^{N} \widehat{\boldsymbol{\alpha}}_{p_k(j)} \, \mathscr{G} \, \widehat{\boldsymbol{\alpha}}_j^{-1} \right), \; \mathscr{I} = \cap_{j=1}^{N} \widehat{\boldsymbol{\alpha}}_j \mathscr{G} \widehat{\boldsymbol{\alpha}}_j^{-1}$$
$$\mathrm{Im} \, \varphi = \mathscr{P} / \mathscr{I} \simeq \sigma_N \leq \mathscr{S}_N.$$

In both cases, the mathematical skeleton is essentially the natural concept of permutations between objects: symmetry operations in crystallography do nothing but swapping equivalent objects with each others.

**Author Contributions:** Conceptual derivation, D.G. and M.Q.; writing–original draft preparation, D.G.; writing–review and editing, M.Q. All authors have read and agreed to the published version of the manuscript.

**Funding:** This research received no external funding beyond the salaries of the authors from CNRS-France.

**Acknowledgments:** The authors are pleased to acknowledge fruitful discussions on bicrystals and dichromatic patterns with Sylvie Lartigue-Korinelk and Richard Portier, and on the general notion of variants with Cyril Cayron.

**Conflicts of Interest:** The authors declare no conflict of interest.

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
