# Peer review of "Bicrystallography and Beyond: Example of Group–Subgroup Phase Transformations"

_crystals, doi:10.3390/cryst10070560_

Round 1
Reviewer 1 Report
This is a sound paper. On one side, it is mathematically rigorous, on the other side it addresses, in fact, to the complex problem of description of integral physical properties of multi-crystalline samples. Although it would be unrealistic expecting its immediate impact, the paper put an order in symmetry-geometrical basement of a crystal-physics problem. I find few positive points in the manuscript. First of all, the authors have chosen very logical and pedagogical way presenting their rather abstract group-theoretical analysis. This later is a general one as it does not limit relative orientation of crystal components of a bi-crystal to any specific lattice sites coincidence. At the same time the authors do not stay on pure mathematical ground describing a physical meaning of some key terms used.
The manuscript is well written and illustrated. It makes an important contribution to the foundation of mathematical crystallography. I have no doubt recommending the manuscript for publication as it is.
Author Response
We wish to thank the referee for his very clear understanding of our work.
Reviewer 2 Report
This is a fine paper. There are only minor "adjustments" to be made, e.g. a space before the quote to other paper at multiple occasions. A few details to be added, e.g. ref [?], 4f and 2f, XXXº, see attached pdf. Also the figures should be better placed, i.e. should only show up in the paper after they have been mentioned in the text.
This study is about a conceptual extension of 3D bicrystallography. It is well known that one can now analyse orthogonal projections of 3D bicrystals with atomic resolution in scanning transmission electron microscopy (STEM), high-resolution phase-contrast parallel-illumination TEM, and various scanning probe methods, e.g. scanning tunneling microscopy. It would be beneficial to mention this fact in the introduction to make this nice paper more relevant to microscopists that study bicrystals and N-crystals in orthogonal projections, i.e. in 2D. The papers https://onlinelibrary.wiley.com/doi/pdf/10.1002/crat.201400071?casa_token=Qo9dXiZRJ_4AAAAA:IJPYkW5WcV_7f9Pcd6V-xkIZ5CQ6gnknM49vSJhqaO1OkfsdftcRJMnMDzYGVirt39eI6uXxPlZKtGc-Qw and https://onlinelibrary.wiley.com/iucr/doi/10.1107/S2053273314022384 could be mentioned in this context.
The authors may also ponder publishing a "practical 2D guid of their approach" for the express benefit of microscopists that study crystalline materials and their grain and domain boundaries. I am sure these researchers would appreciate the conceptual benefit of applying the authors' work in their own investigations.

Author Response
We wish to warmly thank the referee for quoting the remaining typos and his very pertinent remarks in particular refereeing to applications in particular using TEM STEM atom resolution imaging technics.
The two references (Symmetries of migration-related segments of all [001] CSL tilt boundaries in (001) projections for all holohedral cubic materials by P. Moek et al and scanning of 2D space groups by D. Litvin) suggested by the referee are actually a slightly different problem from the one we address: we did not discuss any model of boundaries but only the global symmetry of the abstract construct of "bicrystal" that, of course, does not exist with 3D crystals but is the rational abstract way of determining all the boundaries that are equivalent a given one.
It turns out that the only actual direct possible application of our group derivation is the superimposition of 2D layer structures like graphene : a bicrystal is there a bilayer of twisted graphene sheets that can beautifully be studied by HREM and/or Z contrast BF-DF STEM imaging. This aspect of direct application of our present work is developed in another paper of us entitled "Crystallography of twisted bilayers: example of graphene and similar structures" to be submitted soon to Acta Cryst A. There, the second reference to D. Litvin takes all its sense.
Our point here was to gather, in one coherent paper, the elementary group theoretical aspects of the crystallography of two or more crystals (or quasicrystals/incommensurate structures) independently on the dimension of the implied periodicities and the crystallo-chemical nature of the interfaces. The structure of our paper is unfortunately not suited for introducing a discussion about the interfaces themselves and the various ways of observing them. We fully understand the point of view of the referee and we are very sorry of not being able to add a pertinent short discussion on the subject. But the other paper on graphene layers does.
Reviewer 3 Report
Referee Report
on paper “Bicrystallography and beyond: example of group-subgroup phase transformations “ (crystals-840514) by authors Denis Gratias and Marianne Quiquandon submitted to Crystals
This is interesting paper. It reports about the features of the fine crystal structure and symmetry analysis of the different crystals. This is great fundamental work. The obtained results are interesting and reliable. However, paper needs some improvement only after which it can be accepted. At this stage, my decision is minor revision. But I hope that after revision this paper can be accepted.
- I fully agree with the author that bicrystals are attractive from fundamental point of view But I feel that introduction seems poor. Please add motivation of the study. Why such deep symmetry analysis can be interesting for readers? Please highlight strong features of your research.
- Sorry, But I can’t find references [2,3] in the text. Please revise this carefully.
- I think it will be better if authors will use numbering for all formulas and equations like:
Xxxxxxxx + yyyyyyy = zzzzzz (1)
- Can you add some comments for different papers in which authors provide information about the phase transition between two very similar space groups but with different center of symmetry? I mean when authors described the nature of the ferroelectric properties in quasi-centrosymmetrical systems (DOI: 1016/j.jmmm.2016.10.140; DOI: 10.1016/j.jmmm.2018.05.036).
- I impressed by the paper and hope that it can be published after careful minor revision.
Author Response
1 Indeed we fully agree of having been a bit short in the introduction. We added at the very beginning of the paper the following paragraph:
Understanding how geometric order can propagate in solids, for example with the generation of fascinating Moiré patterns in the superimposition of two or more identical twisted lattices, the specific relative orientations between twinned crystals or the distribution of ordered variants in the morphology of a initial single crystal after a phase transition, is a major part of describing the properties of coherent interfaces in crystals. The subject has met a particular revival since the discovery of bilayers of twisted extended 2D structure like graphene and similar structures where the electronic properties of the bilayer depend crucially on its overall symmetry. Our intention here is to give some general theoretical elements of an answer to this fascinating symmetry problem.
[…]
2 It was a typo error we corrected it; thank you.
3 we did labeled equations but only those that were used later in the paper. Many equations are just for help in the flow of the calculus and do not require being numbered. We think hat it makes the reading a bit easier
4 Indeed The case of the paper DOI: 1016/j.jmmm.2016.10.140 enters perfectly in the present scheme: Transition P63/mmc -> P63cc (same lattice?) where an electron microscopy examination could reveal the existence of inversion boundaries between polar domains with Dark Fields performed images with polar reflexions. Assuming the lattices being identical, we finf the transition being of order 2 like CuZn in which case the permutation group is simply S2. Discussing the the extinction and/or background intensities between adjacent domains is however quite delicate to introduce it in the present context without increasing too substantially the length of the paper. This process applies to any group-subgroup transformation; It has been widely used in the 80’s and 90’s …
5 Thank you very much for your kind and very interesting comments.
Reviewer 4 Report
The manuscript "Bicrystallography and beyond: example of group-subgroup phase transformations" by Denis Gratias and Marianne Quiquandon represents a purely theoretic consideration of generalized crystallography objects produced by superimposing two or more crystals having different mutual orientations. In general, the topic is worth discussing. The manuscript is appropriately structured and authors' conslusions are scientifically sound. But, in my opinion, since the manuscript reports no experimental data it falls well beyond the scope covered by Crystals. I recommend the authors to transfer the manuscript to another more specialized journal, such as Symmetry (https://www.mdpi.com/journal/symmetry/about), which nicely covers mathematical aspects of crystallogrpahy.
Still, I believe the manuscript can be improved by putting some clear associations with experimental crystallography. Could this theory be applied to provide interpretation of HRTEM or SAED patterns in electron microscopy or 2D detector images of reciprocal space in X-ray or electron diffraction. It would be especially interesting for complicated and disputable cases of quasicrystals, incommensurately modulated phases, etc.
The authors consider a couple of prototypical cases of superstructural atomic order in intermetallics, such as CuZn and Cu3Au and classify them according to their theory. Is it possible to make a prediction on other possible types of ordering for the phases with those stoichiometries?
There are few typos in the text.
p. 1, line 14 proper references (presumably 2,3 )should be instead of ?
p. 2. line 56 should read International Tables
Author Response
Thank you for the typos which we have corrected.
The decomposition of a group in cosets of a subgroup has been heavily used in the metallurgy community of order-disorder alloys in the 80’s and 90’s especially for the electron microscopy determination of the observed interfaces using BF-DF imaging technics and LACBED. This applies in fact to any group-subgroup transition in any finite dimension; twins in icosahedral phases lead with 6D group P53M with the unique restriction that a 3D cut by the physical space is needed at the end of the analysis. The two examples given here are just archetypal of this kind of approach and the results depend only on the space-group G and H, not (explicitly) of the stoichiometry.
Our present paper is indeed theoretical with no experimental input. In fact, it was initially written as the “first part” of a two-fold publication to be submitted to Acta Cryst A. The “second part” is now almost achieved and is a long discussion of applying the present material to define the basics of the crystallography of twisted bilayers of graphene and similar 2D structures; it refers directly to the present paper and will be sent soon to Acta Cryst A. The opportunity of sending our first general part to crystals came later when we discovered the project of a special issue devoted to phases transformations that we are pleased to support. We fully understand and accept the point of the referee but we are at a stage where reintroducing experimental inputs would require really major modifications including in our second paper. We are afraid at that stage of the publication, that it is up to the editor choice to withdraw our paper out of the special issue or not; we would fully understand if it is the case.
Round 2
Reviewer 3 Report
Accept as is
Reviewer 4 Report
I'm satisfied with the comments given by the authors and agree with their arguments. I recommend acceptance of the manuscript as is.